# *Plasmodium vivax* infection compromises reticulocyte stability

Martha A. Clark[1], Usheer Kanjee[1], Gabriel W. Rangel [2], Laura Chery[3], Anjali Mascarenhas[4], Edwin Gomes[4], Pradipsinh K. Rathod[3], Carlo Brugnara [5], Marcelo U. Ferreira[6] & Manoj T. Duraisingh [1✉]

The structural integrity of the host red blood cell (RBC) is crucial for propagation of *Plasmodium spp.* during the disease-causing blood stage of malaria infection. To assess the stability of *Plasmodium vivax*-infected reticulocytes, we developed a flow cytometry-based assay to measure osmotic stability within characteristically heterogeneous reticulocyte and *P. vivax*-infected samples. We find that erythroid osmotic stability decreases during erythropoiesis and reticulocyte maturation. Of enucleated RBCs, young reticulocytes which are preferentially infected by *P. vivax*, are the most osmotically stable. *P. vivax* infection however decreases reticulocyte stability to levels close to those of RBC disorders that cause hemolytic anemia, and to a significantly greater degree than *P. falciparum* destabilizes normocytes. Finally, we find that *P. vivax* new permeability pathways contribute to the decreased osmotic stability of infected-reticulocytes. These results reveal a vulnerability of *P. vivax*-infected reticulocytes that could be manipulated to allow in vitro culture and develop novel therapeutics.

[1] Department of Immunology and Infectious Diseases, Harvard T. H. Chan School of Public Health, Boston, MA, USA. [2] Department of Biochemistry and Molecular Biology, Pennsylvania State University, University Park, PA, USA. [3] Department of Chemistry, University of Washington, Seattle, WA, USA. [4] Malaria Evolution in South Asia (MESA)-International Centers of Excellence in Malaria Research (ICEMR), Goa Medical College, Bambolim, Goa, India. [5] Department of Laboratory Medicine, Boston Children's Hospital, Boston, MA, USA. [6] Department of Parasitology, Institute of Biomedical Sciences, University of São Paulo, São Paulo, SP, Brazil. ✉email: mduraisi@hsph.harvard.edu

The unique biconcave shape and structural dynamics of red blood cells (RBCs) permit traversal of the circulatory system and the delivery of oxygen to tissues. RBC structural integrity and deformability is determined by the molecular and cellular functions of the RBC cytoskeleton and membrane transporters[1]. Characterizing RBC stability under osmotic stress is a powerful method for probing the structural properties of the cytoskeleton and membrane transporter activity. An example of the relationship between RBC structural integrity and osmotic stability is seen with naturally occurring polymorphisms in molecules such as PIEZO1 and Ankyrin-1, that disrupt cell volume regulation and cytoskeleton, respectively, and are associated with the clinically relevant RBC phenotypes of hereditary xerocytosis (HX)[2] and hereditary spherocytosis (HS)[3]. These mutations manifest in structurally compromised RBCs with abnormal shapes (stomatocytes and spherocytes), premature hemolysis in vivo[4] and exhibit abnormal sensitivity to hypotonic challenge, which is decreased in HX and increased in HS.

During the *Plasmodium* spp. intra-erythrocytic developmental cycle (IDC), parasite remodeling alters the physiology of the host RBC. In the case of the most well studied of these parasites, *P. falciparum*, changes include disruption of the cytoskeleton by parasite proteins inserted into the RBC membrane[5] and increased plasma membrane permeability[6,7] that culminate in the infected RBC taking on a spherical shape[8]. These changes compromise the structural integrity of the host RBC, and accordingly decrease the osmotic stability of *P. falciparum*-infected RBCs[6,7,9]. The consequence of the reduced stability of *P. falciparum*-infected RBCs is an increased propensity of infected cells to be destroyed in circulation via clearance by the spleen as well as intravascular hemolysis[10].

*Plasmodium vivax*, the most globally widespread of the malaria parasites infecting humans[11], exhibits a strict tropism for reticulocytes[12–14], the youngest of circulating RBCs. This is in contrast to *P. falciparum*, which infects both reticulocytes and mature normocytes. All *P. falciparum* osmotic stability studies have been done in the older, more abundant normocyte fraction. Furthermore, recent observations that *P. vivax* increases reticulocyte deformability[13,15,16], while *P. falciparum* decreases normocyte deformability[5,15,17] suggest that the pathophysiology of *P. vivax* reticulocyte infection is fundamentally different than *P. falciparum* normocyte infection. The absence of an in vitro culture system and appropriate experimental methods have impeded the study of how *P. vivax* impacts the structural stability of the host reticulocyte. Moreover, though reticulocytes[18–20] and erythroid progenitors[21] are known to be more resistant to hypotonic lysis than normocytes, the dynamics of osmotic stability during erythropoiesis and reticulocyte maturation are unknown. Understanding the impact of *P. vivax* infection on the structural integrity of the host reticulocyte has the potential to provide important insight into why a long-term in vitro culture system for *P. vivax* has yet to be established, as well as the in vivo fitness of *P. vivax*-infected reticulocytes.

Osmotic stability is traditionally assessed by measuring release of hemoglobin from bulk RBC preparations in increasingly hypotonic solutions[22]. This methodology cannot resolve the osmotic stability of specific RBC subpopulations within characteristically heterogeneous reticulocyte and *P. vivax* samples. Furthermore, using free hemoglobin as the read-out for RBC osmotic stability is problematic as hemoglobin levels are in flux in both parasite-infected RBCs and uninfected reticulocytes, because *Plasmodium* spp. digest hemoglobin as they mature[23], and reticulocytes are actively synthesizing hemoglobin[24]. Flow cytometry, with its ability to examine discrete cell subsets within heterogeneous populations, has the potential to better define RBC osmotic stability. Finally, though

previous studies have demonstrated the capacity of flow cytometry to detect hemolysis[25–28], the current strategies have not been used to examine the lysis of discrete RBC populations and are furthermore not amendable to collecting sufficient data on rare cell populations.

In this work, we have developed a flow cytometric osmotic stability assay to assess the osmotic stability dynamics within inherently heterogeneous reticulocyte and malaria-infected RBC populations. We observe that osmotic stability steadily decreases during erythroid differentiation and reticulocyte maturation. Of enucleated RBCs, the youngest CD71+ reticulocytes which *P. vivax* preferentially invades using the CD71/transferrin receptor molecule as an invasion ligand[13,14,29], are the most osmotically stable. Examination of clinical *P. vivax* samples showed *P. vivax* infection destabilized the host reticulocyte to levels close to those observed for RBCs from individuals with hemolytic anemia. Furthermore, *P. vivax*-infected reticulocytes are significantly less stable than *P. falciparum*-infected normocytes. Finally, we find that the decreased stability of *P. vivax*-infected reticulocytes corresponded with the appearance of *P. vivax* new permeability pathways (NPPs). The observation that reticulocyte osmotic stability is reduced by *P. vivax* infection suggests that *P. vivax*-infected reticulocytes may be prone to pre-mature hemolysis in vivo. These results reveal a key vulnerability of the parasite that if corrected may facilitate the adaptation of *P. vivax* to in vitro culture, and conversely if targeted could yield novel therapeutics to treat blood stage *P. vivax* infection.

## Results

**Flow cytometry analysis of RBC osmotic stability.** While flow cytometry has been employed to study osmotic stability, previous studies have not harnessed the capacity of flow cytometry to analyze multiple cell populations within a single specimen[25–28]. Therefore, we developed a flow cytometric osmotic stability assay that yields the same hemolysis curves as the traditional hemoglobin absorbance assay and that by quantifying the individual cells that survive hemolysis challenge is able to assess osmotic stability dynamics within heterogeneous RBC populations (Fig. 1a). The analysis of hemolysis by flow cytometry revealed typical RBC forward scatter (FSC) and side scatter (SSC) profiles for non-lytic isotonic conditions, and the subsequent appearance of a low-FSC/low-SSC population in hypotonic, lytic conditions (Fig. 1b and Supplementary Fig. 1).

To confirm that the low-FSC/low-SSC population were RBC ghosts, fluorescent phalloidin, which is excluded from intact cells but binds actin in the cytoskeleton of permeabilized cells was included in the osmotic stability assay. We found that phalloidin stained the FSC-low population that appears in lytic conditions, indicating that the flow lysis assay is sensitive to RBC ghosts (Fig. 1c and d). Considering only intact RBCs, the lysis curves produced by the flow lysis assay were indistinguishable from those of the hemoglobin absorbance lysis assay (Fig. 1e). No difference in the lysis$_{50}$ values (mOsm at which 50% of RBCs lyse) obtained for normal RBCs using the two lysis assays confirmed the accuracy of the flow lysis assay. To test the limits of the flow lysis assay, we measured the osmotic stability of HX RBCs that are resistant to hypotonic lysis. With HX RBCs as well, we observed no difference in lysis$_{50}$ values generated by the two assays (Fig. 1f).

Finally, as *P. vivax* osmotic stability studies included cryopreserved samples and were performed on RBCs maintained in vitro, we examined the effect of (i) cold storage and (ii) *P. vivax* in vitro culture conditions on RBC osmotic stability. As expected[30,31], we observed variation in the osmotic stability of normal RBCs taken from 12 different donors with lysis$_{50}$ values

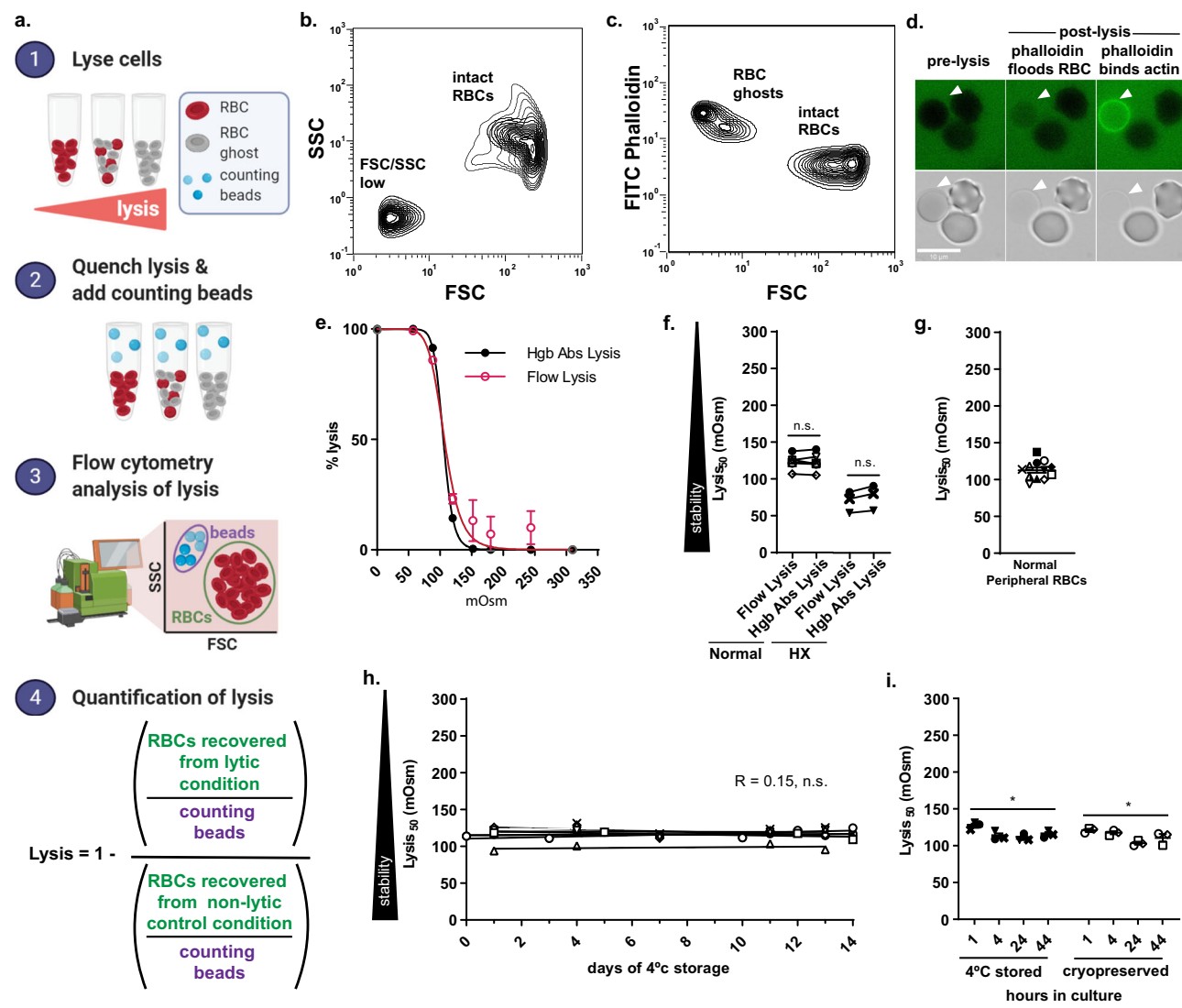

**Fig. 1 Development of a flow cytometry osmotic stability assay. a** Flow cytometry method for measuring RBC osmotic stability. Created with BioRender. com. **b** and **c** Representative flow cytometry forward scatter (FSC) side scatter by (SSC) plot (**b**) and FSC by FITC-phalloidin plot of a lysed (120 mOsm) RBC sample (**c**). Data representative of five independent experiments. **d** Representative immunofluorescent images of FITC–phalloidin binding RBC ghosts. Scale bar, 10 μm. Arrows indicate a RBC undergoing lysis. Data representative of two independent experiments. **e** Representative hemoglobin (Hgb) absorbance (black line) and flow cytometry (red line) lysis curves for normal RBCs. Error bars represent SD of $n = 3$ technical replicates. Data fit with least-squares regression fit curves of normalized data. **f** Normal ($n = 6$) and hereditary xerocytosis (HX) ($n = 3$) RBC lysis$_{50}$ values measured by flow cytometry and hemoglobin absorbance lysis assays. Unique symbols indicate biological replicates and lines match the lysis$_{50}$ values obtained from flow cytometry and hemoglobin absorbance assays. n.s. indicates no significant difference in lysis$_{50}$ (normal RBC $p = 0.8$, HX RBC $p = 0.1$) using paired two-sided Student's $t$-test. **g** Normal RBC ($n = 12$) lysis$_{50}$ values measured by flow cytometry. Unique symbols indicate biological replicates. Horizontal lines and error bars represent mean ± SEM. **h** Normal RBC ($n = 6$) lysis$_{50}$ values over the course of 14 days of 4 °C storage. Unique symbols indicate biological replicates. Data fit with linear regression lines. Mean linear regression for all data depicted by solid line. Pearson correlation of lysis$_{50}$ and days of 4 °C storage, $r = 0.15$. **i** 4 °C degree stored ($n = 3$) and cryopreserved ($n = 3$) RBCs lysis$_{50}$ values following transfer to *P. vivax* in vitro culture conditions. Unique symbols indicate biological replicates. Horizontal lines and error bars represent mean ± SEM. Asterisks denote significant differences in lysis$_{50}$ during culture (4 °C degree stored RBCs $p = 0.02$, Cryopreserved RBCs $p = 0.04$) using RM one-way ANOVA analysis.

ranging from 93.9 to 137.9 mOsm (variation of 11.2%) (Fig. 1g). Subsequent storage of RBCs from six different donors at 4 °C for 2 weeks resulted in no appreciable change (mean slope −0.063 ± 0.24 SEM, Pearson $r = 0.15$) in osmotic stability (Fig. 1h). However, when cryopreserved RBCs and RBCs stored at 4 °C were transferred to *P. vivax* in vitro culture conditions, osmotic stability increased by 13 ±2.6% and 14.6 ± 0.8% after 24 h of incubation (Fig. 1I).

**Osmotic stability decreases during RBC differentiation**. Having established the capacity of flow cytometry to measure RBC

osmotic stability, we next examined the osmotic stability dynamics within the RBC fractions that harbor *P. vivax* infection. To this end, we took advantage of the capacity of flow cytometry to track discrete cell populations, and separately assess the osmotic stability of nucleated RBC precursors, reticulocytes, and normocytes in bone marrow aspirates (Fig. 2a and Supplementary 2A). We found that fluorescent labeling of reticulocytes and nucleated RBC precursors present within bone marrow aspirates allowed us to quantitate the osmotic stability dynamics of each of these RBC subpopulations simultaneously (Fig. 2b). For nucleated RBC precursors we

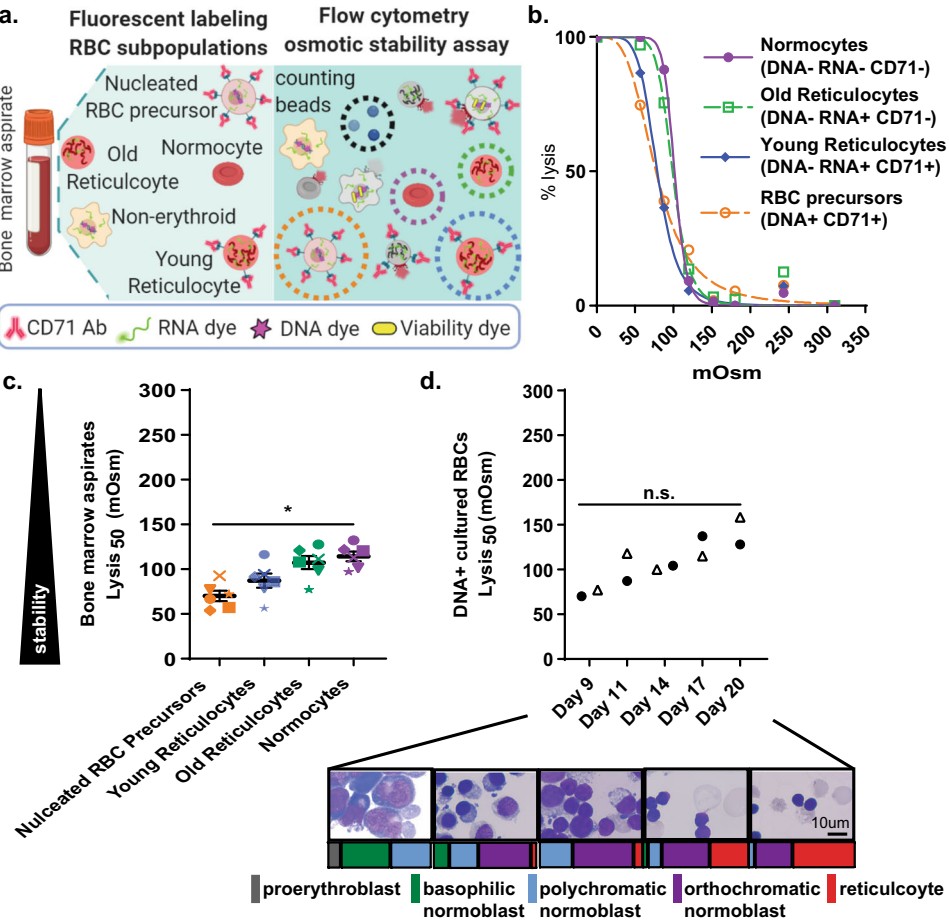

**Fig. 2 Osmotic stability dynamics during erythroid differentiation and reticulocyte maturation. a** Flow cytometry method for measuring osmotic stability of RBC subpopulations. Created with BioRender.com. **b** Representative lysis curves for normocytes (DNA− RNA− CD71−, purple line), old reticulocytes (DNA− RNA+ CD71, green dashed line), young reticulocytes (DNA− RNA+ CD71+, blue solid line), and RBC precursors (DNA+ CD71+, orange dashed line) from a bone marrow aspirate measured by flow cytometry. Data fit with least squares regression fit curves of normalized data. **c** Normocyte (purple data points), old reticulocytes (green data points), young reticulocytes (blue data points), and RBC precursors (orange data points) from bone marrow aspirates lysis$_{50}$ values ($n = 6$). Unique symbol indicate biological replicates. Horizontal lines and error bars represent mean ± SEM. Asterisks denote significant differences in lysis$_{50}$ values ($p = 0.0001$) using RM one-way ANOVA analysis. **d** RBC progenitor lysis$_{50}$ values during course of CD34+ differentiation in vitro ($n = 2$). Unique symbols indicate RBC progenitor lysis$_{50}$ values at days 9, 11, 14, 17, and 20 from two independent differentiations. n.s. indicates no significant difference in lysis$_{50}$ during course of differentiation ($p = 0.2$) using Friedman test. Inset are representative photos of RBC progenitors during in vitro differentiation. Scale bar, 10 μm. Colored bars below images represent proportion of RBC progenitor developmental stages (gray—proeythroblast, green—basophilic normoblast, blue—polychromatic normoblast, purple—orthochromatic normoblast, red—reticulocyte) present at day-9, -11, -14, -17, and -20 of differentiation.

additionally found that lysed cells were also identifiable with a live/dead stain (Supplementary 2B). Importantly, as both osmotic stability studies with bone marrow aspirates and clinical *P. vivax* samples relied on Percoll enrichment to raise reticulocytemia and *P. vivax* parasitemia to reliably measurable levels, we found that Percoll had no effect on the osmotic stability of reticulocytes and nucleated precursors from bone marrow aspirates (Supplementary 2C).

The lysis$_{50}$ values obtained for RBC precursors and reticulocytes from bone marrow aspirates revealed that nucleated RBC precursors (DNA+ CD71+) were the most osmotically stable (Lysis$_{50}$ 70.1 ± 5.8) followed by the youngest CD71+ RNA+ DNA− reticulocytes (Lysis$_{50}$ 87.3 ± 8.0), older CD71− RNA+ DNA− reticulocytes (Lysis$_{50}$ 107.4 ± 7.3) and CD71− RNA− DNA− normocytes (Lysis$_{50}$ 114.5 ± 5.4) (Fig. 2b and c). To establish osmotic stability dynamics during erythropoiesis, we assessed the osmotic stability of erythroid progenitors differentiated in vitro from CD34+ stem cells[19,32]. This study revealed a strong trend of decreasing osmotic stability as cells progressed

in vitro from majority basophilic and polychromatic normoblasts at day 9 (Lysis$_{50}$ 73.8 ± 3.6) to a majority polychromatic and orthochromatic normoblast population at day 11 (Lysis$_{50}$ 102.7 ± 15.6), and then further decreased as cells progressed through final orthochromatic normoblast maturation occurring between day 14 (Lysis$_{50}$ 102.2 ± 2.0) day 17 (Lysis$_{50}$ 126.2 ± 11.2), and day 20 (Lysis$_{50}$ 143.4 ± 15.1) (Fig. 2d). Parallel examination of the osmotic stability of CD71+ reticulocytes differentiated in vitro showed no significant difference in osmotic stability as a function of day reticulocytes were sampled (Supplementary Fig. 2D). Furthermore, consistent with a previous study[19], CD71+ reticulocytes generated in vitro (Lysis$_{50}$ 110.1 ± 3.7) were less stable than CD71+ reticulocytes from bone marrow samples (Lysis$_{50}$ 87.3 ± 8.0) (Supplementary Fig. 2E). Together these results demonstrate the capacity of flow cytometry to assess the osmotic stability of discrete RBC subsets within heterogeneous populations and clearly shows that erythroid development and reticulocyte maturation are associated with significant changes in osmotic stability.

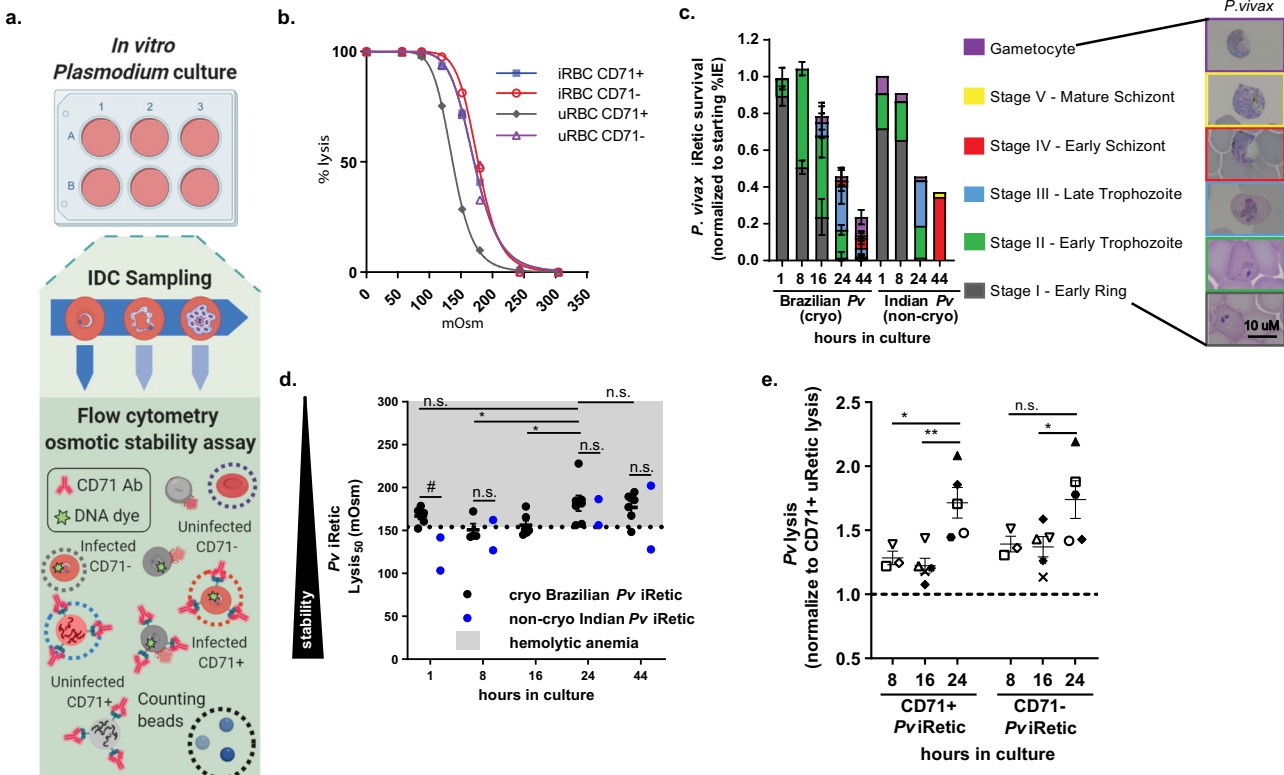

**Fig. 3 Osmotic stability of *P. vivax*-infected reticulocytes. a** Flow cytometry strategy for measuring osmotic stability *Plasmodium*-infected RBCs. Created with BioRender.com. **b** Representative lysis curves for *P. vivax*-infected CD71+ (DNA+ CD71+, blue line) and CD71− reticulocytes (DNA+ CD71−, red line) and uninfected CD71+ reticulocytes (DNA− CD71+, gray line) and normocytes (DNA− CD71−, purple line) from a cryopreserved Brazilian clinical *P. vivax* sample after 1-h in vitro culture. Data fit with least-squares regression fit curves of normalized data. **c** Developmental stage of in vitro cultured ex vivo Indian ($n = 2$) and cryopreserved Brazilian ($n = 5$) *P. vivax* clinical isolates. Horizontal lines and error bars represent mean ± SEM. Inset are representative images of *P. vivax* IDC development stages. Scale bar, 10 μm. Gray (stage I), green (stage II), blue (stage III), red (stage IV), yellow (stage V), purple (gametocyte) bars indicate IDC development stage. **d** *P. vivax*-infected reticulocytes from cryopreserved Brazilian (black symbols) and non-cryopreserved Indian (blue symbols) clinical *P. vivax* samples lysis$_{50}$ values after 1-, 8-, 16-, 24-, and 44-h of in vitro culture. Unique symbols indicate biological replicates. Horizontal lines and error bars represent mean ± SEM. Dashed line (154 mOsm) and gray shading indicate lysis$_{50}$ threshold for hemolytic anemias. Hashtag and n.s., significant and no significant difference in cryopreserved Brazilian and non-cryopreserved Indian *P. vivax* lysis$_{50}$ values, respectively (1-h $p = 0.05$, 8-h $p = 0.6$, 24-h $p = 0.8$, 44-h $p = 1.0$) using unpaired two-tailed Mann–Whitney test. Astericks and n.s., significant and no significant difference between Brazilian *P. vivax*-infected reticulocytes lysis$_{50}$ at 24- and 1-, 8-, 16-, 24- and 44-h of culture, respectively (1-h $p = 0.3$, 8-h $p = 0.02$, 16-h $p = 0.04$, and 44-h $p = 0.9$) using unpaired two-tailed Dunnett's multiple comparisons test. **e** Cryopreserved Brazilian *P. vivax*-infected CD71+ and CD71− lysis$_{50}$ values normalized to lysis$_{50}$ values of uninfected CD71+ reticulocytes at 8- ($n = 3$), 16- ($n = 5$), and 24-h ($n = 5$) of culture. Unique symbols indicate biological replicates. Horizontal lines and error bars represent mean ± SEM. Astericks and n.s., significant and no significant difference between 24-h and 8- and 16-h of culture lysis$_{50}$ values, respectively (CD71+: 8-h $p = 0.04$, 16-h $p = 0.006$ and CD71-: 8-h $p = 0.1$, 16-h $p = 0.05$) using unpaired two-tailed Student's *t*-test.

***P. vivax* infection reduces reticulocyte osmotic stability**. Having established that the youngest CD71+ reticulocytes that are preferred by *P. vivax* for invasion[13,14] are the most osmotically stable of all enucleated RBCs, we next assessed the impact of *P. vivax* infection on reticulocyte osmotic stability. In the absence of continuous *P. vivax* in vitro culture, cryopreserved clinical *P. vivax* samples are an invaluable resource for investigating *P. vivax* biology[33–36]. Cognizant of the decreased stability of cryopreserved RBCs, however (Fig. 1i), we first directly compared the in vitro survival and osmotic stability of cryopreserved and non-cryopreserved *P. vivax* clinical samples as parasites progressed through the IDC (Fig. 3a and b and Supplementary Fig. 3A). Consistent with previous studies[33–36], we observed a 78 ± 3.3% and 63 ± 30.8% loss of *P. vivax*-infected reticulocytes prior to completion of the IDC in vitro (44-h cultures) for Brazilian cryopreserved and Indian non-cryopreserved clinical samples, respectively (Fig. 3c).

Parallel osmotic stability measurements of the uninfected RBCs in *P. vivax* clinical samples revealed that, as observed

previously (Fig. 1I), the stability of cryopreserved uninfected RBC populations increased upon transfer to in vitro culture while non-cryopreserved uninfected RBCs remained steady during the course of culture (Supplementary Fig. 3B and C). For *P. vivax*-infected reticulocytes, we observed that cryopreserved stage I ring-infected reticulocytes were less stable than non-cryopreserved stage I rings. For subsequent time points we observed (i) no difference in the osmotic stability of cryopreserved and non-cryopreserved *P. vivax*-infected reticulocytes and (ii) a decrease in the stability of *P. vivax*-infected reticulocytes as they progressed through the IDC (Fig. 3d). Moreover, when we consider the clinical laboratory cutoff for normal RBCs (lysis$_{50}$ 154 mOsm or 0.45% NaCl)[31,37,38], the osmotic stability of 88% of *P. vivax*-infected reticulocytes (cryopreserved and non-cryopreserved) in 24- and 44-h cultures fell into the range of osmotic instability associated with RBC disorders that cause hemolytic anemia such as HS.

We next examined the degree of instability *P. vivax* induced in the host reticulocyte by comparing the osmotic stability of

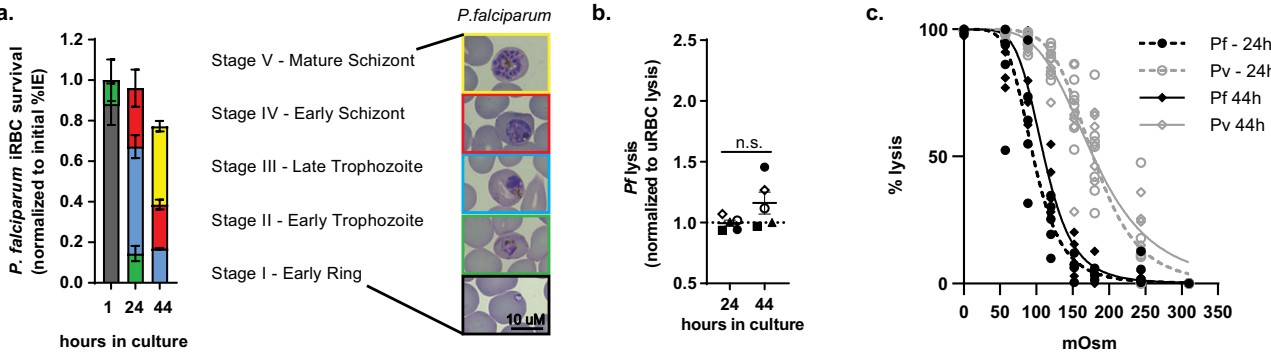

**Fig. 4 Osmotic stability of *P. falciparum*-infected normocytes. a** Developmental stage of in vitro-cultured, cryopreserved *P. falciparum* (clone *3D7* P2G12) ($n = 5$). Bars represent the mean ± SEM. Inset are representative images of *P. falciparum* IDC development stages. Scale bar, 10 µm. Gray (stage I), green (stage II), blue (stage III), red (stage IV), yellow (stage V) bars indicate IDC development stage. **b** *P. falciparum*-infected RBC lysis$_{50}$ values normalized to lysis$_{50}$ values of corresponding uninfected RBCs at 24- and 44-h of in vitro culture of cryopreserved *P. falciparum* (clone *3D7 P2G12*) ($n = 5$). Unique symbols indicate biological replicates. Horizontal lines and error bars represent mean ± SEM. n.s., no significant difference between *P. falciparum*-infected RBC lysis$_{50}$ values at 24 and 44-h of culture ($p = 0.1$) using paired two-tailed Student's *t*-test. **c** Lysis curves of 24-h (gray dotted line) ($n = 7$) and 44-h (gray solid line) ($n = 6$) cryopreserved Brazilian *P. vivax*-infected reticulocyte cultures and 24-h (black dotted line) ($n = 5$) and 44-h (black solid line) ($n = 5$) cryopreserved *P. falciparum*-infected normocytes (clone 3D7 P2G12) cultures. Unique symbols indicate biological replicates. Data fit with least-squares regression fit curves of normalized data.

CD71− and CD71+ *P. vivax*-infected reticulocytes to that of uninfected CD71+ reticulocytes. To take advantage of the availability of cryopreserved *P. vivax* clinical samples while also minimizing the influence of cryopreservation, we excluded the 1-h post thaw time point from subsequent analysis. Due to reticulocyte maturation[39], this analysis was limited to the first 24-h of in vitro culture, as the frequency of CD71+ *P. vivax*-infected and uninfected reticulocytes, respectively, decreased by 56 ± 8.0% and 64 ± 3.5% between 1 and 24-h of culture, and then fell below the limit of detection (0.05% CD71+ reticulocytes) between 24 and 44 h. Of note, the persistence of *P. vivax*-infected CD71+ reticulocytes in our ex vivo cultures through 24 h (Supplementary Fig. 3D and 3E) is longer than previously observed for CD71+ cord blood reticulocytes invaded in vitro by *P. vivax*[13]. The reason for this discrepancy is not immediately evident and therefore subject for future investigation. We found infected reticulocytes were less stable than uninfected CD71+ reticulocytes at all-time points assessed (8-, 16-, and 24-hour cultures), and the progression of *P. vivax* through the IDC further decreased reticulocyte osmotic stability with the appearance of stage III late trophozoite forms in 24-h cultures corresponding to *P. vivax*-infected CD71+ and CD71- reticulocytes being 71.4 ± 12.0% and 74.0 ± 14.6% less stable than uninfected CD71+ reticulocytes (Fig. 3d). Finally, no difference in the stability of CD71+ and CD71− infected-reticulocytes indicated that *P. vivax* infection and not reticulocyte age is the primary determinate of the stability of *P. vivax*-infected reticulocytes (Supplementary Fig. 3F).

**P. vivax-infected reticulocyte are less stable than P. falciparum-infected normocytes.** Considering previous observations that *P. vivax* and *P. falciparum*-infected RBCs exhibit different rheological properties[5,13,15–17], we next compared *P. vivax* destabilization of reticulocytes with *P. falciparum* destabilization of normocytes. To account for the influence of cryopreservation on our *P. vivax* osmotic stability studies, we assessed the in vitro survival and osmotic stability of cryopreserved *P. falciparum* clone *3D7* P2G12[40]. We observed similar progression of *P. vivax* and *P. falciparum* through the asexual IDC but a greater frequency of sexual gametocyte forms for *P. vivax*. In vitro survival, however, was markedly different, with a 77.3 ± 6.5% survival rate observed for *P. falciparum* infected-normocytes at 44-h of culture

compared to a 22.4 ± 0.03 survival rate for *P. vivax*-infected reticulocytes (Figs. 3c and 4a).

We subsequently assessed the osmotic stability of uninfected and *P. falciparum* trophozoite and schizonts stage-infected normocytes in 24- and 44-h cultures (the points at which *P. vivax*-infected reticulocytes were most destabilized). This analysis revealed no difference in the osmotic stability of *P. falciparum* infected (majority trophozoite) and uninfected-normocytes in 24-h cultures and a reduction in *P. falciparum* infected-normocyte (majority schizonts) stability of 23.6 ± 6.9% compared to uninfected normocytes in 44-h cultures (Fig. 4b). This is in contrast to *P. vivax*, which had decreased reticulocyte stability by 74.0 ± 14.6% by the time parasites had matured to the trophozoite form in 24-h cultures. Furthermore, direct comparison of the osmotic stability of *P. falciparum*-infected normocytes and *P. vivax*-infected reticulocytes revealed that *P. vivax*-infected reticulocytes (max Lysis$_{50}$ 184.1 ± 8.3, 24-h culture) are significantly less stable, $p < 0.0003$, than *P. falciparum*-infected normocytes (max Lysis$_{50}$ 107.8 ± 4.7, 44-h culture) (Fig. 4c).

**New permeability pathways increase P. vivax-infected reticulocyte permeability.** New permeability pathways (NPPs) in related malaria parasites, *P. falciparum* and *P. knowlesi*, increase the permeability of the infected RBC to certain solutes[6,7]. To determine whether *P. vivax* possesses NPPs that are contributing to the decreased stability of the host reticulocyte, we assessed the sensitivity of Percoll-enriched cryopreserved Brazilian clinical *P. vivax* samples to the NPP antagonists D-sorbitol and L-alanine. For *P. vivax*-infected reticulocytes, we found that early stage parasites present in 8-h cultures were resistant to D-sorbitol and L-alanine lysis, while the appearance of stage III late trophozoite parasites in 16-h cultures corresponded with *P. vivax*-infected reticulocytes lysing in isotonic D-sorbitol (16-h—47.9 ± 9.0%, 24-h—45.2 ± 9.2%, and 44-h—57.5 ± 7.7%) and L-alanine solutions (16-h—60.7 ± 10.5%, 24-h—61.0 ± 9.2%, and 44-h—55.8 ± 2.3%). Finally, the NPP inhibitor, furosemide[41], protected *P. vivax*-infected reticulocytes from D-sorbitol and L-alanine lysis (Fig. 5a and b). The incomplete lysis of *P. vivax*-infected cells by D-sorbitol and L-alanine along with variation in the sensitivity of different *P. vivax* clinical isolates to D-sorbitol and L-alanine lysis are potentially driven by (i) the high frequency of more stable gametocytes (42.4 ± 9.5% of *P. vivax*-infected reticulocytes in

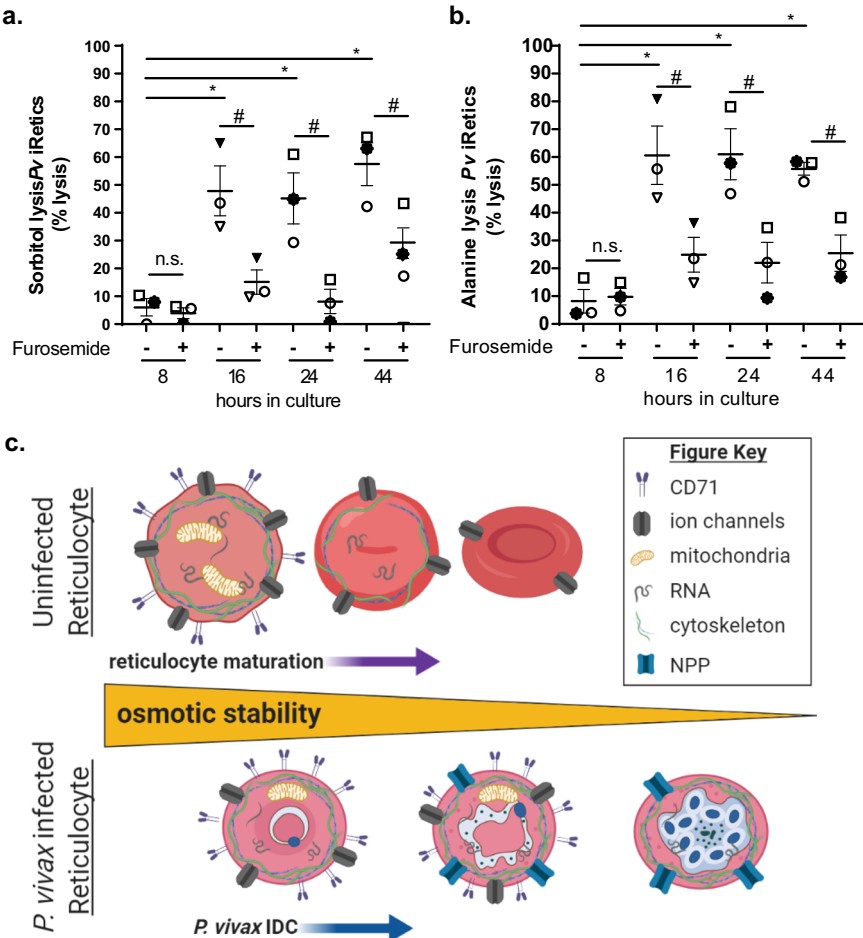

**Fig. 5 _P. vivax_ new permeability pathways increase the permeability of _P. vivax_-infected reticulocytes.** Sensitivity of cryopreserved Brazilian _P. vivax_-infected reticulocytes to **a** D-sorbitol and **b** L-alanine lysis in the presence and absence of NPP inhibitor furosemide in 8-, 16-, 24-, and 44-h cultures ($n = 3$). Unique symbols indicate biological replicates. Horizontal lines and error bars represent mean ± SEM. Hashtag and n.s., significant and no significant difference between furosemide treated and untreated _P. vivax_-infected reticulocyte lysis respectively (sorbitol: 8-h $p = 0.6$, 16-h $p = 0.02$, 24-h $p = 0.04$, 44-h $p = 0.02$ and alanine: 8-h $p = 0.5$, 16-h $p = 0.01$, 24-h $p = 0.03$, 44-h $p = 0.04$) using paired two-tailed Student's _t_-test. Asterisks, significant difference between _P. vivax_-infected reticulocyte lysis at 8-h and 16-, 24- and 44-h of culture (sorbitol: 16-h $p = 0.01$, 24-h $p = 0.02$, 44-h $p = 0.004$ and alanine: 16-h $p = 0.003$, 24-h $p = 0.003$, 44-h $p = 0.005$) using unpaired two-tailed Dunnett's multiple comparisons test. **c** Model of the osmotic stability dynamics within the reticulocyte compartment and the impact of _P. vivax_ infection on reticulocyte osmotic stability. Created with BioRender.com.

44-h cultures)[42,43] (Fig. 3c), (ii) variable NPP activity in different _P. vivax_ isolates, or (iii) _P. vivax_ being less sensitive to D-sorbitol lysis than _P. falciparum_[44,45].

## Discussion

Structural stability of _P. vivax_-infected reticulocytes, critical for the successful propagation of blood-stage malaria infections, has not been thoroughly assessed due to lack of appropriate methodology compatible with _P. vivax_'s unique and challenging biology. To study the impact of _P. vivax_ infection on the structural integrity of the host reticulocyte, we adapted the osmotic stability assay to be analyzed by flow cytometry. This permitted us to work with the limited cell numbers and heterogeneity characteristic of both the reticulocyte compartment to which _P. vivax_ infection is restricted as well as _P. vivax_-infected clinical samples. In the proceeding studies, we observed that the youngest CD71+ reticulocytes, which _P. vivax_ preferentially invades, were the most osmotically stable of enucleated RBC subsets and that _P. vivax_-infected reticulocytes were less stable than uninfected CD71+ reticulocytes. These results indicate that _P. vivax_ severely compromises the structural integrity of the host reticulocyte. In addition to its use in determining the osmotic stability of erythroid precursor and malaria infected-RBC

samples, the single-cell resolution of the flow cytometry osmotic stability assay provides a means for tracking the impact of erythropoiesis perturbations on overall RBC osmotic stability and examining osmotic stability dynamics within heterozygous hematological diseases such as sickle cell and G6PD deficiency.

The decreasing osmotic stability we observed in the later stages of erythropoiesis and then continuing through reticulocyte maturation is likely driven by three major changes occurring to erythroid cells during their differentiation: (i) intracellular solutes changes due to changing membrane transporter abundance and activity[46,47], (ii) cytoskeleton remodeling[48,49], and (iii) the progressive reduction in the surface to volume ratio driven by membrane loss[50]. The observation by us and others[18–20] that reticulocytes are more resistant to osmotic lysis than normocytes is seemingly incongruent with previous reports of membrane instability in reticulocytes[51]. This discrepancy is likely due to either (i) the factors that determine osmotic stability, like higher surface area to volume ratio and greater abundance and activity of membrane transporters in reticulocytes, are more dominant than reticulocyte membrane instability, or (ii) the biology underlying membrane instability and osmotic stability are independent of one another. Ultimately, the factors that determine the stability of erythroid precursors and

reticulocytes are likely multifactorial and highly dynamic. Finally, we and others[19] have found that in vitro differentiated reticulocytes are less stable than ex vivo reticulocytes. Interestingly the addition of cholesterol improved the stability of in vitro cultured reticulocytes[19], suggesting that approaches pursuing the determinants of erythroid progenitor and reticulocyte osmotic stability may yield strategies for generating more stable and consequently more viable RBCs from in vitro cultures.

Our study revealed that *P. vivax* infection decreased the osmotic stability of the host reticulocyte. Though consistent with what has been observed for other *Plasmodium* spp.[6,7,9,52,53], *P. vivax* notably decreased reticulocyte osmotic stability to a significantly greater degree than the more extensively studied *P. falciparum* decreased normocyte osmotic stability. Variation in the instability induced by *Plasmodium* spp. parasites has also been observed for *P. knowlesi*-infected normocytes, that are similarly less stable than *P. falciparum*-infected normocytes[44]. Comparing this data set with ours however, we estimate *P. vivax*-infected reticulocytes are the least stable of the three *Plasmodium* spp., being 70% less stable than *P. falciparum*-infected normocytes whereas *P. knowlesi*-infected normocytes are 30% less stable than *P. falciparum*-infected normocytes. Moreover, *P. vivax*-infected reticulocyte instability is on par with RBC instability observed for hemolytic anemias like HS. Together these findings identify a new parameter of malaria blood stage pathophysiology that differs between *Plasmodium* spp. that has the potential to reveal avenues to species-specific therapeutics.

The observation that *P. vivax* compromises the osmotic stability of the host reticulocyte is consistent with the observation that the majority of late stage *P. vivax*-infected reticulocytes lysed when passed through 2 μm microfluidic channels[16], and suggest that the instability induced by *P. vivax* significantly increases the risk of premature hemolysis of infected reticulocytes. The destabilization of reticulocytes by *P. vivax* infection is also consistent with the precipitous loss of *P. vivax*-infected reticulocytes from in vitro culture observed in this as well as previous studies[33–36]. That said, as *P. vivax* is clearly capable of propagating its blood stage infection in vivo, *P. vivax* likely has strategies for coping with the compromised structural integrity of the host reticulocyte. Future studies will provide important insight into (i) the in vivo consequences of the compromised structural integrity of infected-reticulocytes and how *P. vivax* is able to circumvent the reduced fitness of its host cell to propagate successful blood stage infections and (ii) strategies for stabilizing *P. vivax*-infected reticulocytes in vitro to help facilitate culture adaptation of *P. vivax*.

Intriguingly, clinical observations that *P. vivax* infection results in 4 times higher clearance of uninfected RBCs than P. falciparum[54,55] suggests that *P. vivax* infection not only more severely compromises the host reticulocyte it infects but surrounding RBCs as well. As a consequence of the global destabilization of host RBCs, the frequency and severity of *P. vivax* associated anemia is on par with *P. falciparum*[55–57] associated anemia despite the reticulocyte tropism of *P. vivax* resulting in a lower parasite biomass than *P. falciparum* infections[58]. This anemia is likely the consequence of intravascular hemolysis, opsonophagocytosis driven extravascular hemolysis and dyserythropoiesis[59,60]. The degree to which each of these contributes to *P. vivax* associated anemia remain to be determined.

In our study we found that *P. vivax* NPPs are associated with reduced stability of host reticulocytes. This raises several questions including: (i) are *P. vivax* NPPs required for the acquisition of essential nutrients from the host serum and (ii) what host and or parasite transporters are responsible for the observed NPP activity. The presence of NPP activity in *P. vivax*-infected reticulocytes suggests some overlap between the underlying factors responsible for the reduced osmotic stability of *P. falciparum* and *P. vivax*-infected cells. In *P. falciparum*, Clag3.1 and Clag3.2 have been identified as parasite mediators of NPP activity in infected

RBCs[61,62], and *P. vivax* possesses a *clag* family of genes[63]. This raises the possibility of therapeutically targeting *P. vivax*-infected reticulocytes by inhibiting Clag function, as has been done for *P. falciparum*[64,65].

NPPs are unlikely to be the single variable contributing to the reduced stability of *P. vivax*-infected reticulocytes. Previously reported shedding of microvesicles from *P. vivax*-infected reticulcoytes[66] and 100 nm "holes" in *P. vivax*-infected reticulocytes[13] could each reduce osmotic stability by decreasing surface to volume ratio and increasing permeability of *P. vivax*-infected reticulocytes. Determining the biology underlying the compromised structural integrity of infected reticulocytes has the potential to yield targets against which therapeutics could be developed. Finally, considering the compromised stability of *P. vivax*-infected reticulocytes in the context of the protection from *Plasmodium* infection afforded by RBC polymorphisms such as HS, thalassemia, and G6PD deficiency, it is possible that a mechanism underlying protection from *P. vivax* may be premature hemolysis of infected RBCs.

Together our data support a model in which *P. vivax* referred CD71+ reticulocytes are more osmotically stable than CD71− reticulocytes and normocytes, but, upon *P. vivax* infection and subsequent maturation, the infected reticulocyte undergoes a precipitous loss of osmotic stability. Moreover, the magnitude of instability induced by *P. vivax* is close to the instability observed for hemolytic anemias like HS. Additionally, the onset of NPP activity corresponds with the decreasing osmotic stability of infected reticulocytes (Fig. 5c). As a result, *P. vivax*-infected reticulocytes likely exhibit increased rates of intravascular hemolysis and premature clearance from circulation similar to what is observed for RBCs from individuals with hemolytic anemias. Future studies will focus on the mechanisms underlying these changes in osmotic stability that impact the structural integrity of the host reticulocyte and the consequences of this instability on in vivo survival. Finally, identifying strategies for stabilizing *P. vivax*-infected reticulocyte could prove key to culture-adapting *P. vivax*, while strategies to further destabilize *P. vivax*-infected reticulocytes could conversely yield novel blood stage therapies for *P. vivax*.

## Methods

**Ethics approval**. For HX, bone marrow aspirate, and Brazilian and Indian clinical *P. vivax* samples, we confirm that all relevant ethical regulations were complied with. For HX and human bone marrow samples negative for blasts or diserythropoietic conditions, informed consent was obtained for all patients and samples were collected under the approval of the Boston Children's Hospital Institutional Review Board (IRB04-02-017R) and were de-identified for use in this study. We obtained human IRB waivers for the use of the de-identified parasite samples from the Harvard School of Public Health Office of Human Research Administration for Brazil (IRB21410-101) and for India (IRB17-1071). For Brazilian clinical *P. vivax* samples, informed consent was obtained from all patients. Study protocols for Brazilian parasite sample collection were approved by the IRB of the Institute of Biomedical Sciences, University of São Paulo, Brazil (IRB 1169/ CEPSH, 2014). For Indian *P. vivax* samples, informed consent was obtained from all patients. Study protocols for Indian parasite sample collection were approved by the ethics boards at Goa Medical College and Hospital (GMC) (no number assigned), University of Washington (42271), the Division of Microbiology and Infectious Diseases of the US National Institutes of Health (NIH) (11-0074) and the Government of India Health Ministry Screening Committee (HMSC) for collection of the Indian isolates.

**Parasite sample collection**. Brazilian clinical *P. vivax* samples were collected in the town of Mâncio Lima, Acre State, leukodepleted with BioR 01 Plus filters, cryopreserved with glycerolyte, and stored in liquid nitrogen. The collections were performed in the context of a randomized, open-label clinical trial (NCT02691910). Indian *P. vivax* samples were collected at Goa Medial College and Hospital in Bambolim, Goa, in conjunction with the Malaria Evolution in South Asia International Center of Excellence in Malaria Research and the University of Washington and leukodpleted with CF11.

**Bone marrow and parasite enrichment**. Bone marrow aspirates, cryopreserved Brazilian clinical *P. vivax* samples and *P. falciparum* 3D7 P2G12 were enriched on

1.080 g/mL KCl high Percoll gradients[33]. Briefly, 2 mL of re-suspended cells (up to 25% hematocrit) were layered on 3 mL 1.080 g/mL KCl high Percoll gradient and spun for 15 min at 1200×g. Subsequently, the interface was removed, washed, and applied to assays. For *P. falciparum* 3D7 P2G12 cultures, the interface and pellet were washed and recombined before being applied to assays.

**CD34+ in vitro RBC differentiation**. CD34+ hematopoetic stem cells (HSCs) purchased commercially (Lonza) were differentiated in vitro following the established three-step differentiation protocol[19,32] in Iscove's liquid medium (Biochrom) with 5% human plasma (Octapharma). Giemsa-stained cytospins were prepared and a minimum 200 cells per slide (×1000 magnification) were called as either proerythroblast, basophilic normoblast, polychromatic normoblast, orthochromatic normoblast, or reticulocyte. Images were taken at ×1000 magnification using an Excelis HD Camera attached to an Olympus BX40 microscope.

**P. vivax and P. falciparum in vitro culture**. Indian and Brazilian clinical *P. vivax* samples and *P. falciparum* 3D7 P2G12 were cultured at $100 \times 10^6$ cells per mL in IMDM (Gibco) containing 10% AB+ heat-inactivated sera and 50 µg/mL gentamicin at 37 °C in 5% $CO_2$, 1% $O_2$, and 94% $N_2$[33]. Hemacolor-stained cytospins were prepared and 200 cells per slide (×1000 magnification) were called as either asexual stage (I–V) or gametocyte[67]. Images of cytospins were taken at ×1000 magnification using an Excelis HD Camera attached to an Olympus BX40 microscope. Osmotic stability assays for non-cryopreserved clinical Indian *P. vivax* samples were performed locally at Goa Medical College where samples were collected while cryopreserved clinical Brazilian *P. vivax* samples were shipped from *São Paulo* Brazil to Boston, MA where they were thawed and osmotic stability assays were performed.

**Flow cytometry**. For the described flow cytometry experiments, $1 \times 10^7$ cells were washed with flow buffer prior to staining with combinations of the following fluorophores: Thiazole Orange, 1:1000 (Thermo Fisher Scientific), Vybrant Dye-Cycle Violet, 1:5000 (Thermo Fisher Scientific), Vybrant DyeCycle Green, 1:5000 (Thermo Fisher Scientific), propidium iodide, 1:500 (Thermo Fisher Scientific), α-CD71-APC, 1:25 (Miltenyi Biotech), α-GypA-FITC, 1:100 (Stem Cell Technologies), and FITC-Phalloidin, 1:200 (Thermo Fisher Scientific). Bone marrow and in vitro cultured RBC samples were stained at 4 °C for 20 min and *P. vivax* and *P. falciparum* samples at 37 °C for 20 min. Prior to analysis by flow cytometry, AccuCheck counting beads (Thermo Fisher Scientific) were added to each sample. All flow cytometry experiments were acquired on a Miltenyi MACSQuant instrument equipped with MACSQuantify version 2.11 and 405, 488, and 638-nm lasers and a minimum of 200 events were collected for each cell population being analyzed for the no lysis control sample. All subsequent gating and analysis was based on the no-lysis control sample. As lysis in subsequent samples resulted in ultimate destruction of the cell populations, this minimum event count was not able to be achieved for all samples. For all conditions analyzed, a minimum of 2000 events were collected for cell counting beads. Samples with parasitemia and or reticulocytemia below 0.05% were considered below the limit of detection. Data were analyzed using FlowJo (Version 10.4).

**Osmotic stability assays**. Phosphate buffer saline lysis solutions[22] ranging from 300 to 0 mOsm (distilled water) were made by serial dilution and final osmolarity measured by vapor pressure osmometer (Wescor Vapro 5520). For flow cytometric osmotic stability assays, $1 \times 10^6$ cells (stained with appropriate fluorescent dyes or antibodies) were incubated at $5 \times 10^6$ cells/mL in lysis solutions ranging from ~300 to 0 mOsm (distilled water) for 10 min at 37 °C. Lysis was stopped with 4× volume of quenching solution (1:10 AccuCheck Counting Beads (Thermo Fisher Scientific) to flow buffer). Only cells with normal flow cytometric FSC/SSC profiles (as defined by the control 300 mOsm condition) were considered to be intact. For assays containing nucleated cells, nuclei and dead cells were additionally excluded from the intact cell population. The ratio of beads to intact cells was used to calculate the absolute number of cells remaining in each lytic condition and percent lysis was determined by normalization to the 300 mOsm control condition (0% lysis). For hemoglobin absorbance osmotic stability assays, cells were lysed as described above before 4× volume of flow buffer was added to stop lysis. Cells were pelleted and the absorbance (380, 415, 450, and 540 nm) of supernatants measured on a Spectramax M5 plate reader. The amount of hemoglobin present in supernatants was calculated by the Harboe method[68]. Percent lysis was determined by normalization to the 300 mOsm control condition (0% lysis).

**RBC ghost assay**. For RBC ghost detection by flow cytometry, FITC conjugated phalloidin was added to all lysis and quenching solutions. For RBC ghost detection by immunofluorescence microscopy, RBCs suspended in PBS containing FITC-Phalloidin were allowed to settle on a coverslip before a hypotonic solution (57 mOsm) containing FITC-Phalloidin was added and RBC lysis and ghost formation examined by live video microscopy on a Zeiss AxioObserver.Z1. Images were taken at ×630 magnification on a Zeiss AxioObserver.Z1 inverted fluorescent microscope coupled to an AxioCam MRm camera. Images were processed using Fiji ImageJ 1.52p software.

**Sorbitol and alanine hemolysis assay**. *P. vivax* cultures were incubated at $5 \times 10^6$ cells/mL for 30 min at 37 °C in flow buffer, 280 mM sorbitol with 1% BSA, or 280 mM alanine with 1% BSA in the presence or absence of 100 µM furosemide. Following incubation, cells were washed twice with flow buffer, and stained for flow cytometry analysis.

**Statistics and reproducibility**. Lysis50 values derived from least-squares regression fit curves of normalized data were used to assess the reproducibility of results from biological replicates. Biological replicates shown for (i) normal RBCs, (ii) HX RBCs, (iii) bone marrow aspirates, (iv) CD34+ erythroid culture, and (v) Brazilian and Indian *P. vivax* samples are biologically independent samples, and experiments were performed in batches with 1–3 samples being assessed in a given experiment. For *P. falciparum* 3D7 (P2G12) biological replicates are five independently cryopreserved *P. falciparum* 3D7 (P2G12) clone samples thawed sequentially so that experiments were performed independently. Prism (GraphPad Prism8) and Stata (version 16.0, StataCorp LLC., College Station, TX, USA) were used to test for statistical significance.

**Reporting summary**. Further information on research design is available in the Nature Research Reporting Summary linked to this article.

## Data availability

All data generated or analyzed during this study are included in this published article (and its Supplementary information files). A reporting summary for this article is available as a Supplementary information file. The *P. falciparum* 3D7-P2G12 clone used in this study will be made available upon request. All additional samples were clinical ex vivo samples for which we do not have additional material to supply. Source data are provided with this paper.

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

## Acknowledgements

We thank Juliet Imperial, Flow Cytometry Laboratory, Boston Children's Hospital for providing bone marrow samples and Steven Staffa, MS for providing statistical analysis consultation. This study was supported by U.S. NIH/NHLBI F32 HL136173, NIH R01 AI140751, NIH R01 HL139337, and NIH/NIAID MESA-ICEMR Program Project U19 AI89688, the Canadian Institutes of Health Research Postdoctoral Fellowship, and a Conselho Nacional de Desenvolvimento Científico e Tecnológico (CNPq) research scholarship. The clinical trial that originated samples from Brazil was supported by the Ministry of Health of Brazil and the CNPq, grant 404067/2012-3. The funders had no role in study design, data collection and interpretation, or the decision to submit the work for publication.

## Author contributions

M.A.C., U.K., G.W.R., M.T.D.—Experimental design, execution, and data analysis. L.C., A.M., E.G., P.K.R., C.B., M.U.F.—Clinical management of uninfected and infected blood sample collection, including patient management and ethical clearance. M.A.C., U.K., G.W.R., C.B., M.T.D.—Data interpretation and manuscript writing.

## Competing interests

The authors declare no competing interests.
