## [Peer Review File · Nature Communications]

Reviewer comments, first round:

Reviewer #1 (Remarks to the Author):

Clark et al. Plasmodium vivax infection compromises reticulocyte stability

Usually osmotic stability is done on populations of cells by following hemoglobin release from the population under different osmotic conditions. The present paper developed a method to measure cell stability on individual cells. This flow method allowed the measurement of infected red cells from the population and to measure the effect of infected red cells on stability, including as the parasite develops within the reticulocyte. They found that young reticulocytes are more stable and that *P. vivax* infected erythrocytes are more affected by osmotic stability than *P. falciparum* infected red cells. Why might this matter? One of the goals to grow *P. vivax* in culture and unlike *P. falciparum*, this has failed so far. They raise the possibility that the difficulty could be the reduced stability of *P. vivax* infected red cells compared to *P. falciparum* and suggest the addition of cholesterol to the cells to increase stability.

I have no problems with the science in the paper. A few minor criticisms:

1. Line 287 give a reference to the use of furosemide on the NPP.
2. Line 308: indicating rather than indicate.
3. *P. vivax* has channels that run to the vesicles in the membrane that could affect stability.

Reviewer #2 (Remarks to the Author):

The manuscript by Clark et al describes a novel method to measure the osmotic fragility of different cell populations. The authors used this method to measure the osmotic fragility of cells of the erythroid lineage and found that the younger cell populations were more osmotically stable, and this stability was gradually lost as they mature (somewhat known). Of note were results that CD71+ reticulocytes were more osmotically stable than older CD71- negative reticulocytes and normocytes. In the CD71+ reticulocyte population, the authors indicate that infection with *Plasmodium vivax* (Pv) results in a loss of stability of the infected reticulocytes to levels similar to hemolytic anemias (not previously known), concurrent with the appearance of new permeability pathways. The following are a number of comments/questions that I have on the manuscript.

Major:

- The manuscript is clear and well written, and the methodologies robust and convincing. These experiments are difficult to perform especially for Pv. The findings provide an advancement in our understanding of Pv biology and provide a methodological framework to assess osmotic fragility as discussed. The findings also somewhat provide insight into new therapeutic avenues and could be exploited to improve current short-term Pv culture systems. The visual explanations using cartoons in each figure greatly aid the reader in understanding the experiments. Some of the interpretations and stats could be amended.
- Some of the major Pv conclusions are drawn from Fig 3D. An n=2 from the non-cryopreserved Indian Pv isolates is a concern, particularly with the 2 datapoints being quite distinct, and being fresh experiments representing the 'cleaner' data. Were the experiments on Indian Pv isolates performed locally in India, or transported to a larger lab (this was not clear in the manuscript)? Despite this concern, I can fully understand that obtaining Pv isolates with a high enough parasitemia and enough early rings to perform these assays is very difficult (as well as obtaining large enough volumes following percoll enrichment), which is likely why the Brazilian cryopreserved isolates were introduced. For this reason, I think n=2 fresh Pv isolates is acceptable and the data is particularly important as it highlights that the initial fragility at time zero for the Brazilian isolates is an effect of cryopreservation, which is then recovered 8 hours onwards. However, the presentation of a mean and SEM for n=2 is a bit of a stretch (the data points on their own are probably more appropriate). I see that a student's t-test was used to compare the Brazilian and Indian isolates in Fig 3D, wouldn't a Mann-Whitney test be more appropriate here

considering the data looks non-parametric?

A mean SEM is also used for $n=2$ in Fig 2D and Fig 3C and should be amended as above. For Fig 2D, was a paired one-way ANOVA used (Friedman's test)?

- I cannot determine where the cut-off for hemolytic anemias of 171 mOsm was obtained in Fig 3D apart from a vague statement in the figure legend. Is this a generally accepted cut-off in the context of osmotic fragility?

Also for 1 of the 2 Indian isolates (at each time point) and roughly half of the Brazilian isolates (at each time point except 44 hrs), the lysis₅₀ values are below this cut-off. Therefore, I think the conclusion that Pv reduces osmotic fragility to levels on par with hemolytic anemias could be toned down throughout the manuscript. Perhaps the term 'close to' could be used, or amending to "at least half of the isolates being above the cut off for hemolytic anemias, indicative of"?

- In the context of young reticulocytes, why are the range of lysis₅₀ values in Fig 2 C (<100 mOsm) and Fig 2D (100-150 mOsm) different? Do the values in Supplemental Fig 2D come from these two plots (and for invitro retics, $n=8$ from $n=2 \times 4$ time points)? Are these results likely from the invitro retics being in culture longer, or batch variation? This was not very clear.

- The authors describe the in-vivo relevance of these findings as Pv having an increased risk of intravascular hemolysis. This is likely true, but could this be expanded? It is suggested in the literature that the degree of intravascular hemolysis in Pv is less than Pf, and even less than Pk, yet the authors show that the osmotic fragility of Pv is higher than both other species. Is this explained by a lower number of Pv circulating compared to Pf, plus a minor contribution of iRBC compared to uRBC lysis to the intravascular hemolytic pool?

In addition, is the increased osmotic fragility and likely hemolysis of mature Pv stages an explanation for their disproportionately low levels found in circulation compared to earlier vivax developmental stages?

Minor:

- The flow cytometry gating strategies are important and done well. However, showing contour flow plots can be misleading especially if event numbers are low (which I imagine may be the case for the retics and DNA+ cells). Would it be more appropriate to show density plots? I understand this is a personal preference. The flow cytometry methods could also be expanded. For example, what were the minimum event numbers collected for each sample? Were isotype controls used to aid in gating where possible?

- Some of the data are actually paired samples and in some cases may be more appropriate to show connected lines (eg. Fig 2D) instead of presenting as individual groups with mean SEM. In addition, it seems the majority of statistical tests are for normally distributed data, but with the current sample size shown it may be more appropriate to use non-parametric tests (eg Fig 1F). Where statistical tests are not significant, it is still good to show the p-value instead of 'ns' (again a personal preference).

- R-value in Fig 1H and text is different.

- Fig 3E legend indicates hour 1 is included but data is missing from the figure.

Overall, the Authors provide important insight into Pv biology. The findings raise new questions/hypotheses in the Pv field. The manuscript should be considered for publication following some amendments.

Reviewer #3 (Remarks to the Author):

"Plasmodium vivax infection compromises reticulocyte stability" by Clarke et al
The authors present an insightful work on the structural integrity of reticulocytes infected with P. vivax.

The authors develop a new flow cytometry assay to assess the stability of RBCs. They then use this assay to primarily assess the stability of reticulocytes, reticulocytes infected by *P. vivax*, normocytes infected by *P. falciparum* and to examine the new permeability pathways in *P. vivax* infected RBCs. The main findings are (1) RBCs and reticulocytes become less osmotically stable as they age (2) *P. vivax* prefers to invade the youngest reticulocytes which are the most osmotically stable (3) *P. vivax* infection significantly destabilises the RBC membrane. (4) identification of new permeability pathways that contribute to the decreased osmotic stability of *P. vivax* infected-reticulocytes.

The manuscript is well written and presented. The Figures are clear and the statistical analysis is appropriate.

Specific comments:

Abstract

If the authors think that the finding that reticulocytes and RBCs become more osmotically unstable as they age is important, state it here.

Line 41: Insert the word "that" -- ie.young reticulocytes that are more osmotically stable....

Lines 105-119 This summary of the results seems unnecessarily long for the introduction. Consider cutting.

Results

Line 135 change was to were

Line 135-147 Contains too much detail on the assay development. Consider moving some of the methods section.

General comments:

1) The authors need to make clearer throughout the manuscript and particularly in the intro why studying osmotic stability of *P. vivax* infected RBCs is important for clinical malaria and/or for establishment of a culture system for *P. vivax*.

2) The connection between *P. vivax* preferring to invade the youngest reticulocytes and the osmotic stability of those cells (which is stated in the abstract) is unclear. How do we know this preference is not related to another unmeasured factor or aspect of the young reticulocytes?

3) The authors conclude that reticulocyte osmotic stability is reduced by *P. vivax* infection and that this is a key vulnerability of the parasite.

I am unclear why we cannot conclude from this manuscript that the finding that *P. vivax* decreasing osmotic stability is not a key 'strength' of the parasite rather than a key 'vulnerability'. In fact, the parasite is existing in an unstable RBC and appears perfectly capable of replicating and causing disease in vivo. Can the fact that this is a 'vulnerability' be made clearer please to the non-initiated reader?

4) Since this point is made several times, please explain why it is important to know that *P. vivax* makes the host RBC less stable than *P. falciparum*.

5) Please clarify why and how this work could lead to identification of therapeutic strategies. A further discussion of CLAG3 in *P. falciparum* would potentially help. Maybe also expand on the rationale behind the highly speculative statement made by the authors in Line 329.

.

Reviewer No. 1

1. Line 287 give a reference to the use of furosemide on the NPP. *We thank the reviewer for catching this oversight on our part, a reference for using furosemide as an NPP inhibitor (Kirk et al. JBC 1994) has been added (line 332).*
2. Line 308: indicating rather than indicate. *We thank this reviewer for catching this grammatical error, it has been corrected (line 353).*
3. *P. vivax* has channels that run to the vesicles in the membrane that could affect stability. *The reviewer makes an excellent point that the discussion of potential causes of P. vivax infected reticulocytes instability should be expanded. Discussion of how P. vivax infected-reticulocytes have channels that run to vesicles in the membrane and shed macrovesicles could contribute to their reduced stability has been added to the discussion (lines 445-455).*

Reviewer No. 2

MAJOR

1. Some of the major Pv conclusions are drawn from Fig 3D. An n=2 from the non-cryopreserved Indian Pv isolates is a concern, particularly with the 2 datapoints being quite distinct, and being fresh experiments representing the 'cleaner' data. Were the experiments on Indian Pv isolates performed locally in India, or transported to a larger lab (this was not clear in the manuscript)? *The reviewer notes the absence of an important experimental detail. Experiments with Indian P. vivax isolates were performed locally in India at Goa Medical College where the samples were also collected. This has been clarified in the methods section (lines 518-522).*
2. The presentation of a mean and SEM for n=2 is a bit of a stretch (the data points on their own are probably more appropriate). A mean SEM is used for n=2 in Fig 2D and Fig 3C and should be amended. *We thank the reviewer for carefully reviewing the statistical analysis of our results. In accordance with the reviewers recommendation, mean and SEM have been removed from Figure 2D and Figure 3D.*
3. I see that a student's t-test was used to compare the Brazilian and Indian isolates in Fig 3D, wouldn't a Mann-Whitney test be more appropriate here considering the data looks non-parametric? *We appreciate the reviewer taking into consideration the challenges that comes with working with ex vivo P. vivax samples in the field and suggesting the most appropriate statistical analysis for our limited sample size. As per the reviewer's suggestion, Mann-Whitney test has been used to compare the lysis₅₀ values observed for non-cryopreserved Indian P. vivax samples and cryopreserved Brazilian P. vivax samples and manuscript updated to reflect results of this analysis (line 858). The significance trends with the new analysis remain the same, $p < 0.05$ for 1-hour cultures and $p > 0.6$ for all subsequent time points.*
4. For Fig 2D, was a paired one-way ANOVA used (Friedman's test)? *We again thank the reviewer for recommending more appropriate statistical analysis of these results. An ordinary one-way ANOVA analysis was used initially. As per the reviewer's suggestion, data has been reanalyzed with a Friedman's test and the manuscript updated to reflect the results of this analysis (lines 826). With the new analysis the change in lysis₅₀ values*

is no longer significant ($p > 0.2$). The results section has been revised to reflect this change (line 218).

5. I cannot determine where the cut-off for hemolytic anemias of 171 mOsm was obtained in Fig 3D apart from a vague statement in the figure legend. Is this a generally accepted cut-off in the context of osmotic fragility? *We thank the reviewer for catching this important oversight in the manuscript. References describing the normal range of RBC osmotic fragility (Archer et al. 2014 and Godal et al. 1980 and Medscape) have been added to results section (line 258). Of note, a miscalculation during the initial submission of the manuscript led to our initially reporting 171 mOsm/ 0.5% NaCl as the lysis₅₀ cut off for normal sensitivity of RBCs to osmotic challenge. The value is in fact lower (154 mOsm/ 0.45% NaCl). The manuscript has been revised to reflect this change (lines 257-262).*
6. Also for 1 of the 2 Indian isolates (at each time point) and roughly half of the Brazilian isolates (at each time point except 44 hrs), the lysis₅₀ values are below this cut-off. Therefore, I think the conclusion that Pv reduces osmotic fragility to levels on par with hemolytic anemias could be toned down throughout the manuscript. Perhaps the term 'close to' could be used, or amending to "at least half of the isolates being above the cut off for hemolytic anemias, indicative of....."? *We appreciate the request by the reviewer that we more accurately describe instability of P. vivax infected-reticulocytes. The manuscript has been revised to note what percent of P. vivax infected-reticulocytes in 24 and 44 hour cultures (88%) have lysis₅₀ values greater than the 154 mOsm/ 0.45% NaCl cut off for normal sensitivity of RBCs to osmotic challenge. We have additionally revised the language in the manuscript to more accurately describe the instability of P. vivax infected-reticulocytes (abstract line 47-48, intro line 141-143, results line 257-262 discussion line 397-399).*
7. In the context of young reticulocytes, why are the range of lysis₅₀ values in Fig 2 C (<100 mOsm) and Fig 2D (100-150 mOsm) different? *The reviewers observation that the lysis₅₀ values observed for ex vivo young reticulocytes (<100 mOsm, Fig2C) are lower than those observed for nucleated RBC progenitors differentiating in vitro (100-150 mOsm, Fig2D) is an important one. The lower stability of in vitro differentiated RBC progenitors can at least in part be explained by recent observations by Bernecker et al. that the inherent cholesterol deficiency of RBC in vitro culture conditions results in in vitro differentiated reticulocytes being less stable than ex vivo reticulocytes. This difference in stability of age matched in vitro differentiated and ex vivo CD71+ reticulocytes can also be see in our data (Supplemental Figure 2E). Discussion of this can be found in the results section lines 223-229 and discussion 377-384.*
8. Do the values in Supplemental Fig 2D come from these two plots (and for invitro retics, n=8 from n=2x4 time points)? Are these results likely from the invitro retics being in culture longer, or batch variation? This was not very clear. *The ex vivo bone marrow data shown in Supplemental Fig 2D (Supplemental Fig 2E in the revised manuscript) comes from Fig 3C (young reticulocytes/ DNA- CD71+ RNA+ reticulocytes). The in vitro RBC culture data comes from the newly added Supplemental Fig 2D which shows the lysis₅₀ values for DNA- CD71+ RNA+ reticulocytes obtained at day 11, 14, 17, and 20 of in vitro differentiation. As we do not observe a significant change in the osmotic stability of in vitro differentiated reticulocytes during the time frame in which we took samples (Supplemental Figure 2D in the revised manuscript), suggests the length of in vitro*

culture does not significantly influence in vitro differentiated reticulocyte osmotic stability. That said, with only two bioreps our ability to detect (i) smaller changes in osmotic stability during the course of in vitro differentiation and (ii) batch variation is limited. Finally, to increase the transparency of the data presented in Supplemental Figure 2D (supplemental Figure 2E in the revised manuscript), we have changed the symbols reporting the stability of in vitro differentiated CD71+ reticulocytes in what is now Supplemental Figure 2E to denote the lysis₅₀ values for in vitro differentiated CD71+ reticulocytes sampled at different days of in vitro culture and noted this in the figure legend for Supplemental Figure 2 (line 925-928). In addition a new panel has been added to Supplemental Figure 2 to show the osmotic stability of in vitro cultured CD71+ reticulocytes sampled at different days of culture (Supplemental Figure 2D).

9. The authors describe the in-vivo relevance of these findings as Pv having an increased risk of intravascular hemolysis. This is likely true, but could this be expanded? It is suggested in the literature that the degree of intravascular hemolysis in Pv is less than Pf, and even less than Pk, yet the authors show that the osmotic fragility of Pv is higher than both other species. Is this explained by a lower number of Pv circulating compared to Pf, plus a minor contribution of iRBC compared to uRBC lysis to the intravascular hemolytic pool? In addition, is the increased osmotic fragility and likely hemolysis of mature Pv stages an explanation for their disproportionately low levels found in circulation compared to earlier vivax developmental stages? *The reviewer brings up several very interesting points regarding the potential in vivo implications of our study. Discussion has been expanded to include the points the reviewer introduces here (lines 419-428).*

MINOR

10. The flow cytometry gating strategies are important and done well. However, showing contour flow plots can be misleading especially if event numbers are low (which I imagine may be the case for the retics and DNA+ cells). Would it be more appropriate to show density plots? I understand this is a personal preference. The flow cytometry methods could also be expanded. For example, what were the minimum event numbers collected for each sample? Were isotype controls used to aid in gating where possible? *We appreciate the reviewers request for transparency with regards to the rigor of the flow cytometry analysis performed for this study. Our specific response to each of the reviewers questions and or comments follows here:*
 - a. *To address the reviewers' point that contour plots can be misleading we have included the percent frequency of each indicated population in Supplemental Figures 2A and 3A. We have additionally included additional information in the flow cytometry methods section including the minimum event counts to provide more transparency to the quality of the flow data (lines 536-543). We have however maintained the use of contour plots to more clearly depict the overlaid populations depicted in the flow plots found in the flow cytometry gating strategy figure panels (Supplemental Figures 2A and 3A).*
 - b. *Regarding the use of isotype controls, they were not used in the described studies. The rational for their absence is: (1) staining of erythroid lineage cells with GPA Ab is bimodal and (2) all cell preparations stained with CD71 Ab contained a CD71- normocyte population which served as an internal biological negative control for CD71 AB staining and the CD71 Ab staining patterns that we*

observed are consistent with previously published ones (Malleret et al. PLOSone 2013) Finally, with our gating strategies being able to be set based upon the clear positive and negative staining profiles described above and recent appreciation in the flow cytometry field that isotype controls are not in fact often the ideal negative control for flow cytometry staining due to such issues as differences in the ratio of fluorophore to ab for isotype controls and the ab of interest resulting in differences in fluorescence due to abundance of fluorophore on the ab rather than nonspecific binding of the isotype control, we did not include them in our experiments.

11. Some of the data are actually paired samples and in some cases may be more appropriate to show connected lines (eg. Fig 2D) instead of presenting as individual groups with mean SEM. In addition, it seems the majority of statistical tests are for normally distributed data, but with the current sample size shown it may be more appropriate to use non-parametric tests (eg Fig 1F). Where statistical tests are not significant, it is still good to show the p-value instead of 'ns' (again a personal preference). *We thank the reviewer for the suggestions regarding the presentation of data and reporting of statistical analysis. To note the pairing of samples, we've used unique symbols to indicate paired samples present in Fig 1I, Fig 2D, and Sup Fig 2D. To the reviewers second point about reporting p values (significant and non-significant), the figure legends have been updated include p values for both significant and non-significant results.*
12. R-value in Fig 1H and text is different. *We thank the reviewer for catching this discrepancy, the error has been corrected.*
13. Fig 3E legend indicates hour 1 is included but data is missing from the figure. *We again thank the reviewer for their careful reading and attention to detail, reference to the 1 hour data in Fig 3E legend has been removed as we have focused our analysis on data points not affected by cryopreservation.*

Reviewer No. 3

14. If the authors think that the finding that reticulocytes and RBCs become more osmotically unstable as they age is important, state it here. *We appreciate the reviewers suggestion, the abstract has been revised to more strongly make this point (lines 42-45).*
15. Line 41: Insert the word "that" -- ie.young reticulocytes that are more osmotically stable..... *We thank the reviewer for catching this grammatical error, it has been corrected in the revised manuscript.*
16. Lines 105-119 This summary of the results seems unnecessarily long for the introduction. Consider cutting. *We thank the review for this suggestion, lines 132-146 have been abbreviated, see track changes.*
17. Line 135 change was to were. *We thank the reviewer for catching this grammatical error, it has been corrected in the revised manuscript.*
18. Line 135-147 Contains too much detail on the assay development. Consider moving some of the methods section. *We thank the reviewer for their suggestion, lines 161-163 have been abbreviated, see track changes.*
19. The authors need to make clearer throughout the manuscript and particularly in the intro why studying osmotic stability of P vivax infected RBCs is important for clinical malaria and/or for establishment of a culture system for P vivax. *We appreciate the reviewer's suggestion to emphasize these two important rationales for studying the osmotic stability*

of P. vivax infected-reticulocytes. The introduction (lines 98-101 and lines 146-151) and discussion (lines 404-418 and lines 465-475) have been revised accordingly.

20. The connection between P vivax preferring to invade the youngest reticulocytes and the osmotic stability of those cells (which is stated in the abstract) is unclear. How do we know this preference is not related to another unmeasured factor or aspect of the young reticulocytes? *We appreciate the reviewer pointing out the lack of clarity on this point. The preferential invasion of young CD71+ reticulocytes by P. vivax is attributable to P. vivax use of CD71 as an invasion ligand (Gruszczuk et al. 2019). We have added this detail to the manuscript (lines 138-140) to increase clarity on this point.*
21. The authors conclude that reticulocyte osmotic stability is reduced by P vivax infection and that this is a key vulnerability of the parasite.
I am unclear why we cannot conclude from this manuscript that the finding that P vivax decreasing osmotic stability is not a key ‘strength’ of the parasite rather than a key ‘vulnerability’. In fact, the parasite is existing in an unstable RBC and appears perfectly capable of replicating and causing disease in vivo. Can the fact that this is a ‘vulnerability’ be made clearer please to the non-initiated reader? *We thank the reviewer for this perspective. We view our results as a vulnerability because the host reticulocyte is so clearly destabilized by P. vivax infection, as evidenced by its increased sensitivity to osmotic hemolysis. That said we very much appreciate the reviewers important point that clearly P. vivax is not crippled by the host reticulocytes destabilization as it is still more than capable of perpetuating it’s blood stage infection in vivo. We have acknowledged this point in the discussion and have expanded discussion of how P. vivax could potentially compensate for the compromised stability of the host reticulocyte (lines 412-418).*
22. Since this point is made several times, please explain why it is important to know that P vivax makes the host RBC less stable than P falciparum. *We appreciate the reviewer’s candor on this point. We have made revisions to the introduction (lines 85-93) results (lines 293-294) and discussion (lines 387-402 and lines 420-429) to more clearly communicate the important point that despite being easier to study, P. falciparum is often not a good surrogate for other Plasmodium spp. due to significant differences in the pathophysiology of infection and clinical presentation of disease. Case in point, observations that P. vivax infected-reticulocytes become more deformable while P. falciparum infected-normocytes become less deformable suggested that the rheological impact of P. vivax infection was distinctly different from that of P. falciparum. On the clinical level, the growing body of evidence that P. vivax infection is associated with more frequent and severe episodes of anemia cements a clear difference in the pathophysiology of P. vivax and P. falciparum infection and*
23. Please clarify why and how this work could lead to identification of therapeutic strategies. A further discussion of CLAG3 in P falciparum would potentially help. Maybe also expand on the rationale behind the highly speculative statement made by the authors in Line 329. *We thank the reviewer for pointing out the lack of thorough discussion of these two ideas. The discussion section has been revised to more clearly and thoroughly explain the idea that (1) seeking to increase the stability of in vitro differentiated reticulocytes could yield more viable reticulocytes (lines 378-385) and (2) developing drugs that target the determinants of P. vivax infected-reticulocyte stability could yield strategies for treating blood stage infection (lines 439-441).*

Reviewer comments, second round:

Reviewer #2 (Remarks to the Author):

The authors have addressed all comments from the first round of reviews appropriately and have significantly improved the manuscript. I have no further comments/suggestions and would like to congratulate the authors on this excellent paper.

Reviewer #3 (Remarks to the Author):

Thank you for responding to all of my queries and comments.

We thank the reviewers for their time and the invaluable feedback they provided. The manuscript has been improved as a result of these reviews.

REVIEWERS' COMMENTS

Reviewer #2 (Remarks to the Author):

The authors have addressed all comments from the first round of reviews appropriately and have significantly improved the manuscript. I have no further comments/suggestions and would like to congratulate the authors on this excellent paper.

Reviewer #3 (Remarks to the Author):

Thank you for responding to all of my queries and comments.